# Predicting Erectile Dysfunction after Highly Conformal, Hypofractionated Radiotherapy to the Prostate

Kevin Martell [1,*], Conrad Bayley [1], Sarah Quirk [2], Jeremy Braun [1], Lingyue Sun [1], Wendy Smith [3], Harvey Quon [1] and Kundan Thind [4,*]

1   Department of Oncology, University of Calgary, Calgary, AB T2N 4N2, Canada
2   Harvard Medical School, Harvard University, Boston, MA 02115, USA
3   Varian Medical Systems, Calgary, AB T2N 4N2, Canada
4   Henry Ford Health, Detroit, MI 48202, USA
*   Correspondence: kjmartel@ucalgary.ca (K.M.); kthind1@hfhs.org (K.T.)

**Simple Summary:** Erectile dysfunction is a common side effect after any treatment for prostate cancer. This study worked to identify whether worsening erectile function after moderately fractionated radiotherapy could be predicted based on several factors including radiotherapy plan dosimetry. Previous evidence of erectile dysfunction and the mean dose of radiation received by the penile shaft both predicted worsening erectile dysfunction after radiotherapy.

**Abstract:** Background: Erectile dysfunction (ED) is common after prostate cancer treatment. It has been studied for conventional radiotherapy, but associations in the hypofractionated radiotherapy context are less clear. This study aimed to determine which factors are predicted for worsening ED after highly conformal, modestly hypofractionated radiotherapy to the prostate. Methods: Two hundred and twelve patients treated with 6000 cGy in twenty fractions across four centers were included in this study. Demographic, clinical, and dosimetry factors were then evaluated for post-treatment declines in erectile function using logistic regression and an explainable machine learning-based neural network. Results: 212 patients with a median follow-up of 3.6 years were evaluated. A total of 104 (49%) patients received androgen deprivation therapy. Prior to treatment, 52 (25%) patients were on ED medication. Mean doses to the penile bulb, penile crus, and penile shaft were 2490 (IQR: 1529–3656) cGy, 2095 (1306–3036) cGy, and 444 (313–650) cGy, respectively. Fifty-nine (28%) patients had a worsening of ED after treatment. On multivariable analysis, only the mean dose to the penile shaft [OR >345 vs. ≤345: 4.47 (1.43–13.99); $p = 0.010$] and pretreatment use of ED medication [OR yes vs. no: 12.5 (5.7–27.5); $p < 0.001$)] predicted for worsening ED. The neural network confirmed that the penile shaft mean dose and pre-treatment ED medication use are the most important factors in predicting ED. Conclusions: Pre-treatment ED and penile shaft dosimetry are important predictors for ED after hypofractionated radiotherapy for prostate cancer.

**Keywords:** prostate cancer; hypofractionation; EBRT; erectile dysfunction; dosimetry

## 1. Introduction

Patients with intermediate-risk prostate cancer have a variety of treatment options available to them, such as surgery, brachytherapy, and external beam radiotherapy (EBRT). As overall survival outcomes between these regimens are generally considered similar, patient preference and differences in the side effect profile of each treatment play a significant role in the decision-making process [1–5].

A well-established priority for many patients receiving treatment for prostate cancer is the preservation of erectile function [6–8]. In terms of radiotherapy treatment, a relationship has been established between the dose of radiation planned to the penile bulb and the development of erectile dysfunction (ED) for conventional fractionation regimens utilizing 3D conformal techniques [9–11]. However, modestly hypofractionated radiotherapy

treatments of 6000 cGy in 20 fractions have been gaining popularity since the publication of large randomized controlled trials which demonstrated acceptable efficacy [12–14]. For these regimens, the incidence of ED has been approximated at 34% and further study is ongoing [15]. Associations between the penile shaft (glans penis), penile crus, penile bulb dosimetry, and erectile functional outcomes are less established, but a single analysis suggests a relationship does exist [16].

Given the scarcity of current data on ED and moderately hypofractionated treatment regimens, the present study aims to review whether the penile bulb dosimetry and the penile shaft (including the penile crus/penile cavernosum) dosimetry were associated with new diagnoses of ED, or worsening ED in patients receiving moderately hypofractionated EBRT as a primary treatment for prostate cancer.

## 2. Materials and Methods

With the health research ethics board approval (HREBA.CC-21-0502), the electronic medical records and radiotherapy plans for all patients from four centers (two academic quaternary cancer centers and two regional satellite cancer centers) who received 6000 cGy in 20 fractions as a definitive local treatment for prostate adenocarcinoma were retrospectively reviewed. Then, the centralized electronic health record and pharmaceutical information network that covers the entire healthcare jurisdiction was interrogated to determine if any prescriptions for medications for ED were filled by these patients. Patients were included in the analysis if they received 6000 cGy in 20 fractions to the prostate with or without proximal seminal vesicles as part of their curative-intent treatment between 2016 and 2019. Patients were excluded if they had a concurrent diagnosis of another pelvic malignancy, had received prior treatment for prostate cancer or had received prior radiotherapy to the pelvis.

Various radiotherapy planning practices were assumed across the four centers. Variability in prostate radiotherapy planning practices has previously been established across these centers [17]. Over 20 different radiation oncologists with varying levels of contouring expertise and subspecialization of practice provided care over the time period. All plans were dosimetrist-created, volumetric modulated arc therapy plans generated using the Eclipse treatment planning system (Varian Inc, Palo Alto, CA, USA). Doses were calculated using the analytical anisotropic algorithm (AAA). Whether the penile bulb or penile shaft was contoured was physician dependent. Dose volume constraints followed those of a large randomized phase III trial of patients receiving moderately hypofractionated radiotherapy of 6000 cGy in 20 fractions to the prostate [12]. As per the trial, no specific constraints on the penile bulb or genital structures were used at the time of planning at any center, unless the structure was contoured, and a constraint was specified by the treating radiation oncologist. This was an uncommon occurrence (<20% of cases). All plans were delivered using daily cone-beam CT-based image-guided radiotherapy. When androgen deprivation therapy (AD) was combined with radiotherapy, standard practice was to use 2–3 months of neoadjuvant ADT, followed by 3–4 months of concurrent and adjuvant ADT to a total of 6 months ADT duration.

To ensure contour uniformity, the penile bulb, penile crus and penile shaft structures were retrospectively contoured or re-contoured by a single radiation oncologist with a primary practice in genitourinary malignancies, and doses were recalculated retrospectively. The penile bulb structure is defined and contoured as the portion of the bulbous spongiosum of the penis immediately inferior to the GU diaphragm, but without extension to the shaft or pendulous portion of the penis as seen on CT imaging [18]. To ensure the simplicity and reproducibility of contouring, the penile shaft (glans penis) structure was uniformly contoured as the entire penis excluding the penile bulb. This included the corpus cavernosum, corpus spongiosum, and penile shaft as visualized on CT imaging. An example of radiotherapy structure contouring is provided in Figure 1.

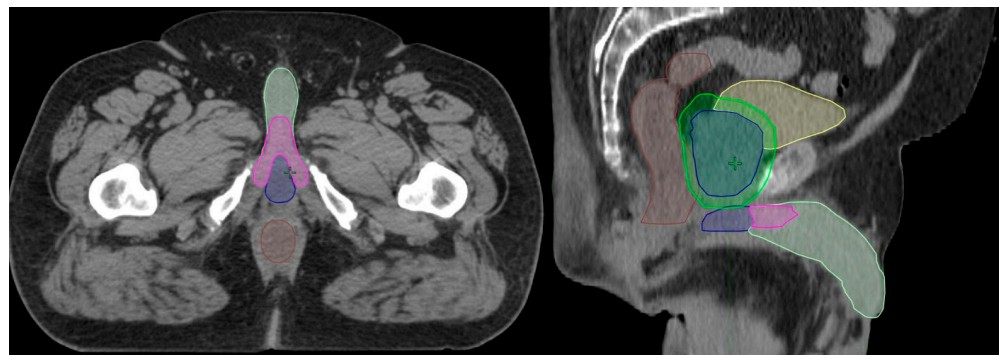

**Figure 1.** Example contours of the penile bulb (blue), penile crus (magenta), and penile shaft (light green) on axial and sagittal CT imaging. Other structures viewed in the image include the rectum (brown), bladder (yellow), prostate (blue), and PTV (green).

When considering how to evaluate worsening erectile function, because of the multi-provider care model utilized across the study institutions, and inconsistencies in documentation practices, medical record documentation of ED was deemed insufficient to ensure that all cases were captured. To capture the ground truth, the pharmacy information network databases were interrogated to determine if any study patient filled a prescription of any phosphodiesterase-5 inhibitor or hormonal treatment (e.g., intracavernosal testosterone injection), as these are uniformly used as first management steps in patients without contraindications. The electronic health records for these patients were then reviewed to ensure no other treatments were employed. Patients were deemed to have developed worsening ED if they started ED treatment after completing treatment with EBRT or ADT (whichever end date was later). Patients were also deemed to have worsening ED if they used an increased dose of ED medication, or had any other intervention for ED after completing their treatment with EBRT or ADT.

For statistical analyses, the Shapiro–Wilk test of normality was used to determine normality in all variables. Descriptive statistics were used to describe the cohort. Normally distributed variables were described using the mean and standard deviation, and non-normally distributed variables were described using the median and interquartile range (IQR). For binomial and ordinal variables, absolute counts and percentages were used. Logistic regression analyses with covariates of hormone therapy use and patient age were used to determine if the mean dose to the penile bulb or penile shaft dose-volume constraints were predictive of ED toxicity. A Youden based area under the receiver-operator curve analysis was performed on the penile bulb and penile shaft dose-volume statistics to determine if a specific cutpoint constraint(s) best predicted for ED toxicity. The cutpoint(s) found were then used as a binomial variable and the logistic regression analysis was repeated. For machine learning-based approaches, a single training dataset was used. First, a principal component analysis was performed on all the data collected on this patient population, with a request to maintain the nine features most helpful in predicting a worsening of erectile function after radiotherapy. A neural network was created, and explainable machine learning techniques of permutation feature importance plots, shapely plots, and accumulated local effects plots were used to enhance the comprehension of the algorithm. All data were analyzed using the r-programming language version 4.0.0 (www.r-project.org).

## 3. Results

### 3.1. Clinical Description of the Cohort

Two hundred and twelve patients with a median follow-up of 3.6 (3.2–4.4) years were identified. Median prostate volume was 36 (27–47) cc on ultrasound imaging at the time of trans-rectal prostate biospy. The median time from biopsy to initiation of radiotherapy was 4.6 (3.3–6.6) months. Median pre-treatment PSA was 9.6 (7.2–13.4) ng/mL. Fourteen (7%)

patients had Gleason grade group 1, one hundred and thirty-six (64%) had grade group 2, fifty-two (25%) had grade group 3, and nine (4%) had grade group 4 or 5 disease. One hundred and thirty-nine (66%) had T1, sixty-six (31%) had T2, and seven (3%) were treated for T3 or T4 disease.

Information on medical comorbidities is presented in Table 1. A total of 130 (61%) patients had a diagnosis of hypertension, with 62 (29%) being on a beta blocker. Fifty-three (25%) patients had a history for smoking at least a twenty pack a year. In total, 52 (25%) patients were on medication for ED prior to radiotherapy. Of these, 20 (38%) were on sildenafil, 28 (54%) were on tadalafil, and 4 (8%) were on vardenafil.

**Table 1.** Description of clinical comorbidities with known associations with erectile dysfunction for a cohort of 212 patients receiving moderately hypofractionated radiotherapy to the prostate.

|  | Number (%) or Median (IQR) |
| --- | --- |
| Age [years] | 72 (67–76) |
| Anxiety | 8 (4%) |
| Beta Blocker | 62 (29%) |
| Chronic Obstructive Pulmonary Disease | 28 (13%) |
| Coronary Artery Disease | 57 (27%) |
| Depression | 12 (6%) |
| Diabetes | 63 (30%) |
| Dyslipidemia | 98 (46%) |
| Hypertension | 130 (61%) |
| Hypogonadism | 0 (0%) |
| Obesity | 19 (9%) |
| Peripheral Vascular Disease | 6 (3%) |
| Stroke | 9 (4%) |
| Transient Ischemic Attack | 12 (6%) |
| Smoking history |  |
| never smoker | 115 (56%) |
| quit > 2 years ago | 53 (26%) |
| current smoker | 37 (18%) |
| Drinking History (any lifetime use) |  |
| 0–7 drinks/week | 173 (84%) |
| 7–14 drinks/week | 16 (8%) |
| >15 drinks/week | 16 (8%) |

### 3.2. Dosimetric Description of Cohort

All patients (100%) completed the 6000 cGy in 20 fractions course of radiotherapy. The median RT duration was 0.9 (0.9–1.0) months. In total, 104 (49%) received androgen deprivation therapy with their course of radiotherapy. The relevant tested dosimetric parameters are presented in Table 2. The median (IQR) values for the mean dose to the penile bulb, penile crus and penile shaft were 2490 (1529–3656) cGy, 2095 (1306–3036) cGy and 444 (313–650) cGy, respectively.

**Table 2.** Dosimetric outcomes for a cohort of 212 patients receiving moderately hypofractionated radiotherapy to the prostate.

|  | Median (IQR) |
| --- | --- |
| CTV volume [cc] | 51.7 (38.9–65.9) |
| PTV volume [cc] | 158.9 (129.8–192.1) |
| PTV V95 [%] | 99.8 (99.5–99.9) |
| Conformity Index | 1.1 (1.1–1.2) |
| Gradient Index [95%–50%] | 3.1 (3.0–3.2) |
| Penile bulb volume [cc] | 4.7 (3.6–6.2) |
| Penile bulb mean dose [cGy] | 2490 (1529–3656) |
| Penile bulb V1000 cGy [%] | 73.5 (49–96.6) |

**Table 2.** *Cont.*

|  | Median (IQR) |
| --- | --- |
| Penile bulb V2000 cGy [%] | 49.1 (24.4–70.9) |
| Penile bulb V3000 cGy [%] | 34.9 (13.6–59.7) |
| Penile bulb V4000 cGy [%] | 22.6 (6.1–46.4) |
| Penile bulb V5000 cGy [%] | 11.8 (0.3–35.5) |
| Penile bulb V6000 cGy [%] | 1.6 (0–13.7) |
| Penile crus volume [cc] | 6.5 (5.1–8.5) |
| Penile crus mean dose [cGy] | 2095 (1306–3036) |
| Penile crus V1000 cGy [%] | 79.3 (49.9–98) |
| Penile crus V2000 cGy [%] | 42.6 (18.6–73.1) |
| Penile crus V3000 cGy [%] | 22 (4.5–47.9) |
| Penile crus V4000 cGy [%] | 9.6 (0.1–27.1) |
| Penile crus V5000 cGy [%] | 2.1 (0–11.3) |
| Penile crus V6000 cGy [%] | 0 (0–1) |
| Penile shaft volume [cc] | 93.3 (80.6–106.2) |
| Penile shaft mean dose [cGy] | 444 (313–650) |
| Penile shaft V1000 cGy [%] | 11.9 (6.9–19.8) |
| Penile shaft V2000 cGy [%] | 4.4 (1.8–8.6) |
| Penile shaft V3000 cGy [%] | 2 (0.4–4.1) |
| Penile shaft V4000 cGy [%] | 0.8 (0.1–2.1) |
| Penile shaft V5000 cGy [%] | 0.2 (0–0.9) |
| Penile shaft V6000 cGy [%] | 0 (0–0.1) |

### 3.3. Evaluation of Erectile Function Outcomes

Forty-nine (23%) patients were on medications for ED after treatment. Among this cohort, 24 (49%), 23 (47%) and 2 (4%) received prescriptions for sildenafil, tadalafil and vardenafil, respectively post-radiotherapy. No (0%) patient had an increase in their prescribed dose of pre-medication for ED after radiotherapy. On closer evaluation, 25 (12%) patients received a new prescription for ED medication after radiotherapy, and 28 (13%) had their ED medication stopped. After evaluating for any perceived changes in ED medication use [53 (25%)], or procedures relating to ED [6 (3%)], 59 (28%) patients were considered to have had a worsening of erectile function after radiotherapy treatment.

On initial logistic regression, the mean dose to penile bulb [OR 3662 vs. 1546: 1.42 (0.91–2.20); $p = 0.122$] did not predict for ED. Trends towards the mean dose to the penile shaft [OR 634 vs. 310: 1.36 (0.96–1.92); $p = 0.081$], and the mean dose to the penile crus [OR 2966 vs. 1293: 1.50 (0.97–2.32); $p = 0.068$] being predictive of worsening ED were observed. Of the other tested predictors, only a pre-treatment prescription for ED medication predicted for worsening function after radiotherapy [OR yes vs. no: 11.73 (5.61–24.51); $p < 0.001$]. Notably, age [OR 76 vs. 67: 0.88 (0.58–1.32); $p = 0.54$], and the utilization of androgen deprivation therapy [OR yes vs. no: 0.77 (0.41–1.43); $p = 0.411$] were not associated with the outcome.

On Youden cutpoint analysis, the cutpoint for the mean dose to the penile bulb was 2630 cGy [area under ROC curve (AUC): 0.571]. The cutpoint for the mean dose to the penile shaft was 345 cGy [AUC: 0.615], and the cutpoint obtained for the mean dose to the penile crus was 1726 cGy [AUC: 0.595] (Figure 2).

Using these cutpoints, a repeated logistic regression univariate modeling indicated that the mean dose to the penile bulb [OR > 2630 vs. ≤2630: 1.81 (0.97–3.37); $p = 0.062$] did not predict for ED outcomes. Cutpoints for the mean dose to the penile shaft [OR > 345 vs. ≤345: 4.19 (1.85–9.48); $p = 0.006$], and the mean dose to the penile crus [OR > 1726 vs. ≤1726: 2.41 (1.21–4.79); $p = 0.012$] both appeared to be associated.

On multivariabe analysis, using the three cutpoints obtained and the clinical features explored above, only the mean dose to the penile shaft [OR > 345 vs. ≤345: 4.47 (1.43–13.99); $p = 0.010$] and the pretreatment use of ED medication [OR yes vs. no: 12.5 (5.7–27.5; $p < 0.001$)] predicted for worsening ED.

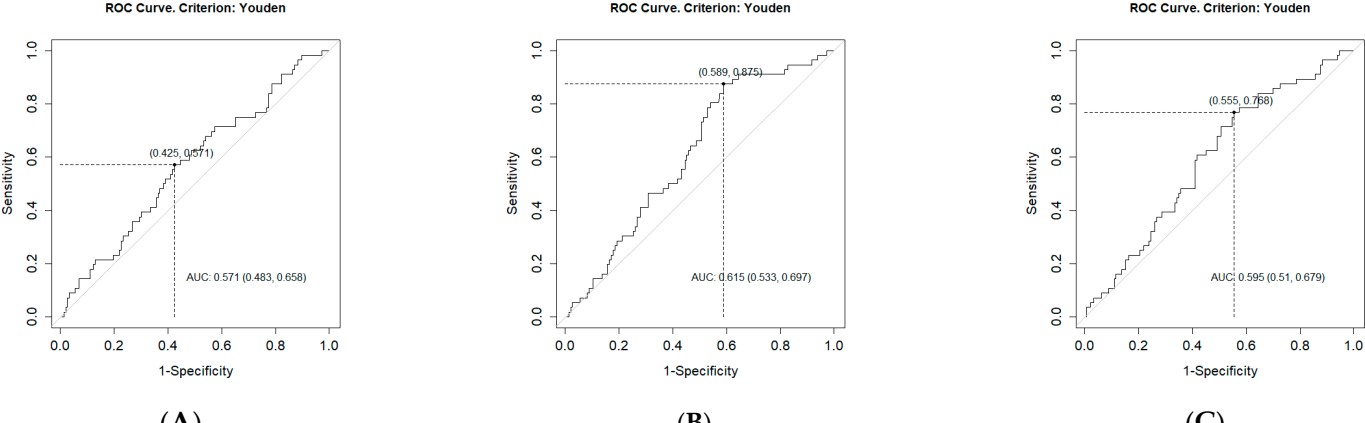

**Figure 2.** Receiver curve analysis for cutpoints in predicting clinical outcomes of worsening erectile function after modestly hypofractionated radiotherapy to the prostate. Plots are as follows: (**A**) mean dose to the penile bulb, (**B**) mean dose to the penile shaft, and (**C**) mean dose to the penile crus.

### 3.4. Machine Learning-Based Analysis of Erectile Function Outcomes

A neural network-based explainable machine learning approach was also applied to the cohort to review the predictors for worsening ED after radiotherapy. After an initial principal component analysis requesting the nine most important factors be preserved, this again identified pre-treatment prescriptions for ED medication as the most important predictor of post-treatment worsening of erectile function. Interestingly, none of the other clinical factors were retained in this principal component analysis. The permutation feature importance plot of the final nine factors is presented in Figure 3, and the accumulated local effects plots for each of these factors is presented in Figure 4.

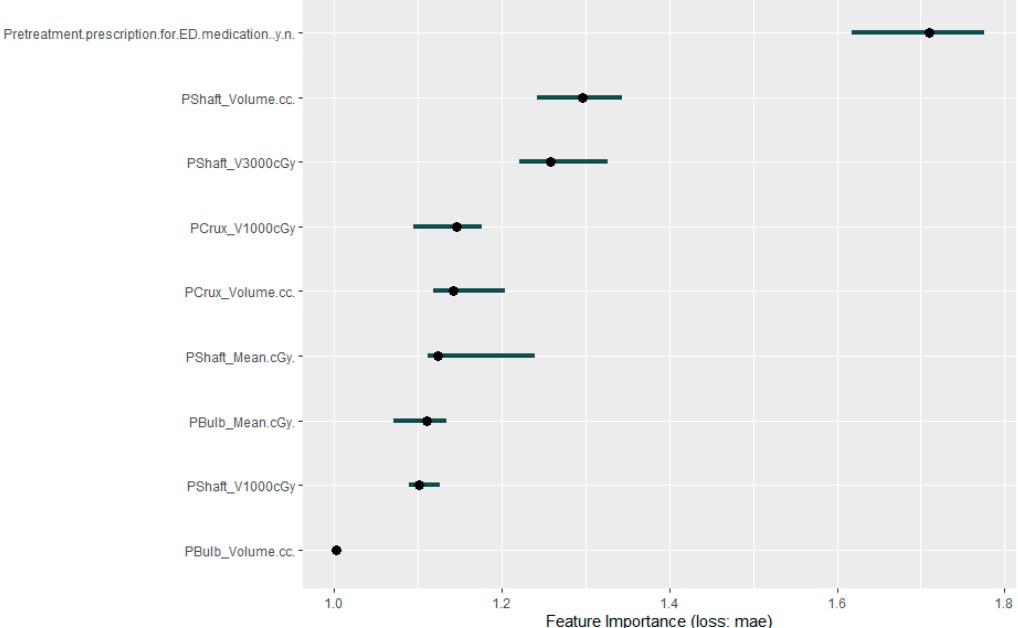

**Figure 3.** Permutation feature importance plot for the 9 factors considered as most predictive of worsening erectile dysfunction after moderately hypofractionated radiotherapy to the prostate.

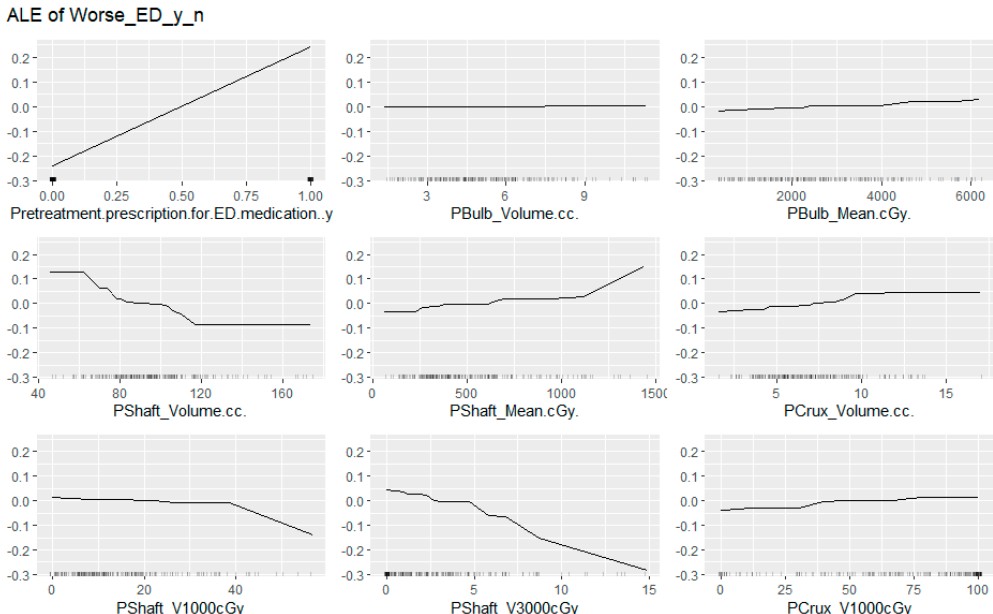

**Figure 4.** Accumulated local effects plots for the 9 factors considered as most predictive of worsening erectile dysfunction after moderately hypofractionated radiotherapy to the prostate.

## 4. Discussion

This study identified medication use for treatment of ED prior to moderately hypofractionated radiotherapy to the prostate as the most important predictor of subsequent worsening of erectile function. The most predictive dosimetric factor appeared to be the mean dose to the penile shaft on regression analysis with a mean dose >345 cGy predicting for worse outcome.

The finding that the penile shaft dose better predicted for erectile function outcomes than other dosimetric parameters is best informed by the machine learning algorithm. A review of factor importance and accumulated local effects demonstrated an inverse relationship between the penile shaft volume and the probability for worsening of erectile function after radiotherapy. The mean dose to the penile shaft essentially verifies two important contributors to the overall risks of ED: the gland size and the dose deposition to the genitals. This evidence for an effect from both the overall volume and dose is supported by the plots indicating the percentage of structure receiving both 1000 cGy and 3000 cGy. There is a relative threshold tolerance for 40% of the penile shaft receiving 1000 cGy, but a sharp falloff with any more than 5% receiving 3000 cGy. In general, the conformality of the plans reviewed was very good (Conformity index of 1.1), resulting in only a very small volume of the penile shaft structure receiving an in-field radiation dose, and most accumulated low radiation dose was out of field. There was much less variation within the penile bulb and penile crus structure volumes. This is likely attributed to the less pronounced effect of their dosimetric correlates on both logistic regression and neural network-based analyses.

When considering the functions of the penile bulb, penile shaft and penile crus as an explanation for why the penile shaft may be the strongest predictor of erectile dysfunction, one should note that the primary function of the bulbospongiosis muscle is to increase the force behind ejaculation. The ischiocavernosus muscle is primarily responsible for trapping blood within the crus and penile shaft. The penile shaft itself, as contoured in this study, included the penile crus, corpus cavernosa, corpus spongiosum, and the glans penis. These structures constitute the majority of the vasculature responsible for maintaining an erection, in addition to the majority of innervation that allows for sensation. Given this, from a functional and anatomical basis alone, a significant argument can be made that irradiation of this structure should have an impact on erectile dysfunction. The results of

the present study suggest there is potential of fibrosis of this vasculature at even low doses of radiotherapy.

This study demonstrated that in this cohort of patients, well-established clinical risk factors for ED, namely age and medical comorbidities, had minimal impact on the outcomes overall [19,20]. Although this is not immediately intuitive, this study was purely focused on the incidence of worsening erectile function over a relatively small portion of each patient's lifespan (approximately five percent). Of note, in this cohort, over the median follow-up, only approximately three percent of patients would be expected to develop erectile dysfunction should no intervention have been applied [21]. Therefore, patients with a predisposition to ED due to medical comorbidities, including age, likely experienced ED prior to radiotherapy. Patients with a weakened erectile function prior to radiotherapy were likely either on medication prior to radiotherapy, had already received more aggressive interventions, or were not interested in treatment for ED. Within this context, pre-radiotherapy treatment for ED is indeed the best predictor possible for the clinically relevant worsening of erectile function after medication, as it encompasses the relevant clinical features.

Additionally, there was a lack of an association between androgen deprivation therapy use and erectile function outcomes. There are several possible explanations which would require further analyses that are out of the scope of the present study. For instance, it is well known that genitalia become smaller from androgen deprivation therapy [22–24]. The practice varied across centers and providers, but a typical approach to the timing between androgen deprivation therapy and radiotherapy would involve using androgen deprivation therapy for 2–3 months prior to CT simulation imaging for radiotherapy planning. This would provide enough time for the penile shaft to shrink from the androgen deprivation therapy. Hence, androgen deprivation therapy use may as well be accounted for in the structure volume variables used in this study. Otherwise, there are possibilities that androgen deprivation therapy was used for more advanced disease, which would be seen more frequently in older patients with a baseline-worse erectile function, or possibilities that there was a high degree of a lack of recovery of libido due to sub-optimal testosterone recovery, and little desire to seek out new medical management for new ED [25,26]. However, given the relatively short duration of ADT overall (six months), one should not interpret this to apply to longer overall durations of ADT.

Within the present literature, the analysis by Murray et al. provides the only known guidance on considerations for dosimetric outcomes and ED after moderately hypofractionated radiotherapy to the prostate [16]. In this re-analysis of the Chhip data, they found a cutpoint mean dose to the penile bulb of >20 Gy and advanced age predicted for worsening ED outcomes. Although there are differences in study design, the analyses are relatively comparable. The cutpoint obtained for penile bulb dosimetry in the present dataset (2630 cGy) was not statistically significant as a predictor ($p = 0.06$). This may be due to a difference in the outcome being measured (RMH EP scores versus clinically relevant ED defined by a change in the use of ED medication or intervention for ED). However, penile bulb dosimetry was still present in the neural network model, and likely does have utility in predicting ED despite a significant degree of conflicting literature on the topic [27]. The dosimetric marker found to be a superior predictor in the present study (penile shaft/glans penis) was not evaluated in the Murray study [16].

There are a number of strengths and limitations to the present analysis. Notably, all data for any prescription filled or procedure performed in the health jurisdiction was available for analysis. Hence the only missing data would be for patients who received the treatment and subsequently moved to a different region (essentially a move further than 500 km away) after radiotherapy. The objectivity of reviewing an outcome such as changes in management for ED allows this study to circumvent issues that would be seen with clinician reporting, patient tolerance of treatment, or patient bother by the clinical ED outcomes. However, an inherent fault would be patients who are concerned about ED after radiotherapy, but have not sought out medical advice regarding it, would not

be captured in the present analysis. This should be mitigated somewhat, as in informal discussions with them, a majority of the radiation oncologists in the health jurisdiction endorsed that they deliberately will ask patients about erectile function after treatment, even if that information is not deliberately captured in their follow-up notes. Another limitation to this study design is the potential for patients to have been counted as having had worsening erectile dysfunction if they in fact had an improvement in function and went from using ED medication to no use of ED medication after treatment, or perhaps tried ED medication before radiotherapy but were found unresponsive and did not fill subsequent scripts. Fortunately, these scenarios would be uncommon and are unlikely to impact the results of the present analysis.

With regards to the dosimetry presented in the present study, it is not possible to confirm that the penile position was constant throughout the treatment course. However, given the dose contribution to the penile shaft was mainly from scatter dose, it is likely that differences in the position had little impact on the overall dosimetry received by the penile shaft through the treatment course. Nonetheless, there are multiple known limitations to retrospective analyses such as this one, including the risks of inherent bias, or unknown confounders. The prevalence of inherent bias with the use of machine learning algorithms is potentially lower however, as such algorithms should provide a more objective approach to synthesizing the data to answer the question asked of it. However, they would still be prone to biases emerging from which data are available. A deliberate attempt was made to capture as much information as reasonable to avoid potential unknown confounders within this study; however, in the absence of deliberate randomization of patients to either high or low doses of radiation to the genitalia (which would be unethical), this potential can never be fully addressed.

Despite these possible limitations, the present analysis does provide sufficient evidence for a clinical practice change to encompass dosimetric evaluations of genitalia, specifically the penile shaft/glans penis structure in the context of pre-existing use of medication for ED treatment. This will better enable radiation oncologists to counsel patients on their individual risks of ED following their treatment.

## 5. Conclusions

Pre-treatment ED and penile shaft dosimetry are important predictors for ED after hypofractionated radiotherapy for prostate cancer. Attention should be paid to these structures when planning radiotherapy cases. This new finding, if validated, may in the future help physicians afford patients more specific estimates of risks of erectile dysfunction after receiving radiotherapy.

**Author Contributions:** Conceptualization, K.M. and K.T.; methodology, K.M., K.T. and S.Q.; software, S.Q. and J.B.; validation, K.M., S.Q., K.T. and C.B.; formal analysis, K.M. and K.T.; investigation, K.M. and K.T.; resources, K.M., J.B., L.S., W.S. and H.Q.; data curation, C.B. and K.M.; writing—original draft preparation, K.M.; writing—review and editing, K.M., K.T. and S.Q.; visualization, K.M.; supervision, K.M. and K.T.; project administration, C.B. and K.M. All authors have read and agreed to the published version of the manuscript.

**Funding:** This research received no external funding.

**Institutional Review Board Statement:** The study was conducted in accordance with the Declaration of Helsinki and approved by the Health Research Ethics Board of Alberta (HREBA.CC-21-0502, December 2022).

**Informed Consent Statement:** A waiver of individually signed informed consent was provided by the Health Research Ethics Board of Alberta as data were accessed retrospectively and the study was deemed to be minimal risk compared to the societal benefit derived within its design (HREBA.CC-21-0502).

**Data Availability Statement:** Data are not made publicly available due to ethical restrictions on patient information.

**Conflicts of Interest:** The authors declare no conflict of interest.

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
