# Peer review of "Predicting Erectile Dysfunction after Highly Conformal, Hypofractionated Radiotherapy to the Prostate"

_radiation, doi:10.3390/radiation3020008_

Round 1
Reviewer 1 Report
This is a study using a neural network to determine the risk of erectile dysfunction after radiotherapy to the prostate and the surrounding structures. This is an important topic to investigate in the era of increased value of QoL after specific cancer treatments. The paper is well structured and well written. Although the authors should be congratulated with this paper, there is one important issue that should be addressed further:
The primary outcome is erectile dysfunction as represented by start or increase of ED-medication after treatment. This is of course a very rough and incomplete way of measuring ED. 12% stopped ED-medication after treatment, were these patients cured of their ED by the radiotherapy? It is unclear what the state of erectile dysfunction was before treatment (patients without medication could have had an ED already) and it remains unclear how much more severe the ED became after treatment (maybe they were not interested in trying out medication). They could also have ordered medication online. Therefore, I think this major flaw should be described better in the discussion section. Also, I would suggest changing the aim to evaluate clinically relevant ED (since these were the patients who actively asked for medication) instead of ED in general.
The concept that penile shaft dose influences ED is new. I suggest adding to the discussion section, what the theory behind this could be.
Author Response
Thank you for your kind comments. A primary assumption of this study was that any cessation of medication for erectile dysfunction would be due to a worsening of either interest or function. This assumption does leave a potential flaw in that if patients had improvements in erectile function or were “just trying” it without having significant erectile dysfunction before the treatment or for patients who had trialed ED medication and then had no response prior to treatment would be included as having a worsening of function. Unfortunately there was no conceivable way to mitigate this given the data available. We believe our conservative approach of assuming any cessation of ED medication that was used just prior to radiotherapy was due to a worsening function. We have added this concept to the limitations. Otherwise, we completely agree with your remaining comments. Ideally, this study will have to be confirmed with prospective survey or questionnaire based assessments of ED over time in a separate cohort of patients. This leads to significant limitations as outlined in the discussion.
We have added some of the anatomical justifications as to why our results might be. We agree it is new and exciting for its potential impact on men’s health.
Reviewer 2 Report
In the article "Predicting erectile dysfunction after highly conformal, hypofractionated radiotherapy to the prostate" Martell et al. approach a neglected topic of the toxicity of radiotherapy treatment in prostate cancer: erectile dysfunction. With a focus on the new schemes of hypofractionated radiotherapy, the article offers a novel concept associating clinical and therapeutic characteristics with dosimetric data as possible predictors of erectile dysfunction. 3 anatomical structures (penile bulb, penile shaft and penile crus) are delineated and evaluated as possible structures at risk. Pre-treatment ED medication is identified as an essential factor in the prediction of ED. What is noteworthy is the use of an artificial intelligence algorithm with neural networks to identify ED predictors. Comparing the study methodology with the inclusion criteria of other studies, the authors correctly identify possible risk confounding bias. As a minor observation I recommend to reeplace "shaf" with "shaft" in the abstract. The authors include tables with clinical and dosimetric data, but also a permutation importance plot and an example of contouring of penile structures. Correctly identifying the possible effect of antiandrogenic therapy, the authors do not mention anything about the reproducibility of the position of the penis during the radiotherapy treatment. Would contention systems be necessary to limit the positional variations of the penis between fractions? I would also mention radiobiological data about alpha beta ratio and the adaptation of dosimetric constraints to the converted dose in "2GY". Also, due to the lack of specific expertise in neural network, I would recommend the evaluation of the manuscript by an expert in this field.
Author Response
Thank you for your comments. With regards to the reproducibility of the penile position we recognize this as a potential confounder to our trial. Specifically, we are unable to confirm if penile position changed during each treatment. However, our expectation is that given the relatively low dose to that structure overall being mainly from internal or external scatter that position should not have a large impact on the overall dosimetry actually received by the penile shaft. We have added this comment in our discussion.
As a group we did not feel converting the dosimetry to equivalent dose in 2Gy fractions would be helpful to the radiation oncology community as a whole and may complicate our message. Specifically, both the PROFIT and CHHIP trials argued the radiobiological rationale for hypofractionation in prostate cancer with an assumed a/b of 1.5 for prostate tissue(s) and have been published and widely adopted with appropriate clinical protocols implemented in many centers for evaluating rectal and bladder dose constraints based on 20 fraction regimens. Our hope with the present study was to present the community with a constraint for erectile function based on an assumption of a 20 fraction regimen. As the dose constraints calculated in the paper are based on this it helps avoid confusion and having the wider readership make equivalent dose calculations on their own to be able to use this.
Reviewer 3 Report
This study retrospectively investigated predictive factors for erectile dysfuction after moderately hypofractionated radiotherapy. Their results suggesting the importance of the mean dose to the penile shaft and pretreatment ED medication may be of some interest. However, one of the major drawbacks of this study is the lack of detailed analysis regarding the influence of androgen deprivation therapy (ADT). Although the authors stated in Discussion that ADT was not associated with ED development, results of such analyses are not described in the Results section at all. Details of ADT are also not stated in Materials and Methods. How long was ADT performed, and how did ADT not influence ED occurrence?
Evaluation ED from the ED medication before and after treatment is objective, but may not reflect the true incidence of ED. 13% of the patients stopped to receive ED medication after treatment; in this population, it is unclear whether the stoppage is due to the loss of patient desire or improvement of the erectile function.
Furthermore, the authors failed to discuss the medical reasons for the association of the penile shaft dose with ED. Why was the penile shaft dose more important than the doses to the penile bulb or crus? It should be discussed with respect to the functional role of the penile shaft, bulb, and crus.
Specific points
Line 37: “available to them” is repeated in the same sentence.
Line 121-124: Is this sentence correct?
Author Response
Specific points
Line 37: “available to them” is repeated in the same sentence.
Thank you.
Line 121-124: Is this sentence correct?
Corrected thank you.
Thank you for your comments. When ADT was used it was used for a duration of 6 months consistently as per the CHHIP trial’s publication. We have added this to the methods section.
With regards to ADT not being a predictive factor for ED outcomes. We recognize this was not heavily emphasized but we would direct your attention to section 3.3 of the results for the logistic regression outcomes. Here we have identified ADT as being notable for NOT having an association with ED outcomes therapy [OR yes vs no: 0.77 (0.41-1.43); p=0.411].
Otherwise, in our discussion we provide a paragraph on our interpretation as to why the ADT would perhaps be included in other variables and lead to being a non-statistically significant predictor for ED outcomes (paragraph 4). We recognize that a lack of association between ADT use and ED goes against conventional understanding of ED.
Considering your comment about the penile shaft volume being the most important. As seen on the permutation feature importance plot the volumes of all three structures were seen as predictive. We completely agree that penile shaft being the most important is a novel finding. This is why we feel this study warrants publication and further exploration. There’s a multitude of different possible explanations for why one of these structures might have been more important than the others but unfortunately without getting into more nuanced substructure based analyses (which would likely need MR linac based contouring) we are really unable to comment on the exact mechanism or which substructure is most important. However, we recognize this is likely one of the most important criticisms of this study and the functions of each of these structures contributes to our findings. Given this we have added a new paragraph (between 2 and 3) in the discussion.
Round 2
Reviewer 1 Report
The authors responded well to the comment.
Reviewer 3 Report
Adequately revised